**Subject Category:**
Biology (whole organism)

ecology

detection probability, density, home range, *Plethodon cinereus*, red-backed salamander, spatial capture–recapture

**Author for correspondence:**
Raisa Hernández-Pacheco
e-mail: rai.hernandezpacheco@csulb.edu

# Unexpected spatial population ecology of a widespread terrestrial salamander near its southern range edge

Raisa Hernández-Pacheco[1], Chris Sutherland[2], Lily M. Thompson[1] and Kristine L. Grayson[1]

[1]Department of Biology, University of Richmond, Richmond, VA, USA
[2]Department of Environmental Conservation, University of Massachusetts, Amherst, MA, USA

  RH-P, 0000-0002-3681-5127; CS, 0000-0003-2073-1751; LMT, 0000-0002-3821-0324; KLG, 0000-0003-1710-0457

Under the current amphibian biodiversity crisis, common species provide an opportunity to measure population dynamics across a wide range of environmental conditions while examining the processes that determine abundance and structure geographical ranges. Studying species at their range limits also provides a window for understanding the dynamics expected in future environments under increasing climate change and human modification. We quantified patterns of seasonal activity, density and space use in the eastern red-backed salamander (*Plethodon cinereus*) near its southern range edge and compare the spatial ecology of this population to previous findings from the core of their range. This southern population shows the expected phenology of surface activity based on temperature limitations in warmer climates, yet maintains unexpectedly high densities and large home ranges during the active season. Our study suggests that ecological factors known to strongly affect amphibian populations (e.g. warm temperature and forest fragmentation) do not necessarily constrain this southern population. Our study highlights the utility of studying a common amphibian as a model system for investigating population processes in environments under strong selective pressure.

## 1. Introduction

Many species of amphibians have experienced severe population declines in recent decades [1–5], and these global trends are expected to continue under increasing rates of climate change and anthropogenic disturbance [6–8]. To ameliorate these threats,

major conservation efforts have focused on maintaining species richness, especially targeting rare or endemic species near extinction [9]. Yet, common species arguably have larger impacts on ecosystem structure and function due to their substantial biomass and capacity for energy turnover [10–14]. Widespread common species also provide several compelling advantages for understanding population responses to changing environments; their high abundances overcome sample size limitations for accurately quantifying population processes, and their large geographical ranges allow studying these processes across environmental gradients, while potentially exhibiting similar sensitivities to environmental change as imperilled species [11]. Within the context of the current global biodiversity crisis where amphibians lead the list of threatened terrestrial species [15], it is critical that we increase our general understanding of population processes in common amphibians to better predict responses to global environmental change and mitigate ecosystem disturbances [16].

The advantages presented by widespread common species are further highlighted in range edge habitats at low latitudes where populations are subjected to environmental conditions that may resemble those of future populations facing warmer landscapes. Environmental conditions in these areas are characterized by warmer temperatures [17], a factor expected to negatively affect the demography of amphibian populations. For instance, warmer temperatures have been shown to severely limit amphibian populations by constraining their growth and survival [18], as well as recruitment [19]. Consequently, low-latitude range edge populations in general are expected to be at lower abundances than central or continuous populations across the range and thus may be more susceptible to stochasticity and higher risk of local extinction [20].

Growing evidence has accumulated that even common amphibian species are undergoing dramatic population changes. For instance, some populations of plethodontids (lungless salamanders), which are known for their relatively high abundance in streams and forest floor communities and key roles in food webs and nutrient cycling, have been reported to be declining in mountainous regions potentially due to pathogens [21] or increased niche overlap and displacement due to climate-related shifts in range [22,23]. However, many of these studies have been conducted on populations within large protected areas that serve as salamander biodiversity hotspots in North America (e.g. Appalachian Mountains) and probably experienced a different evolutionary history to those at range edges. Range edge populations may have novel adaptations that increase their fitness under environmental pressure [20,24,25], e.g. increased body size in warmer environments [26]. Thus, understanding patterns of abundance at range limits, together with knowledge on how individuals distribute their activities in space and time, will contribute to a strong evidence-based framework for amphibian management and conservation in a rapidly changing world.

Here, we use newly developed models of spatial population processes to investigate patterns of seasonal activity, density and space use of a widespread amphibian, the eastern red-backed salamander (*Plethodon cinereus*), in a suburban forest near its southern range edge. This species is tightly linked to the productivity of forests, with significant contributions to both salamander biomass (estimated over 90% in some populations) and overall vertebrate biomass (estimated to be twice that of birds and equal to small mammals [27,28]). It is also a species that is highly sensitive to warm temperatures and habitat loss [29–32]. For example, their surface activity is known to be influenced by variations in temperature and rainfall [18] and climate-modelled distributions predict a contraction of its geographical range and other plethodontids under future warmer climates [33,34]. Under these conditions, we expected low densities for *P. cinereus* near its southern range limit. Given that population density and home range size often vary inversely potentially due to resource limitations [35]—a hypothesis yet to be tested among amphibians [36], we also expected this population to exhibit a relatively small home range size. Contrary to this, our analysis reveals extremely high densities and relatively large home ranges in a suburban southern population of *P. cinereus*, highlighting the value of this species as a model for understanding amphibian population ecology.

# 2. Methods

## 2.1. Study species

The eastern red-backed salamander (*Plethodon cinereus*) is a common and widely distributed lungless woodland salamander, ranging from Quebec and the Maritime Provinces in Canada in the north, to North Carolina in the south and westward to Minnesota, USA ([37]; figure 1). *Plethodon cinereus* is associated with cool climates, temperate forests and high moisture habitats as it relies solely on

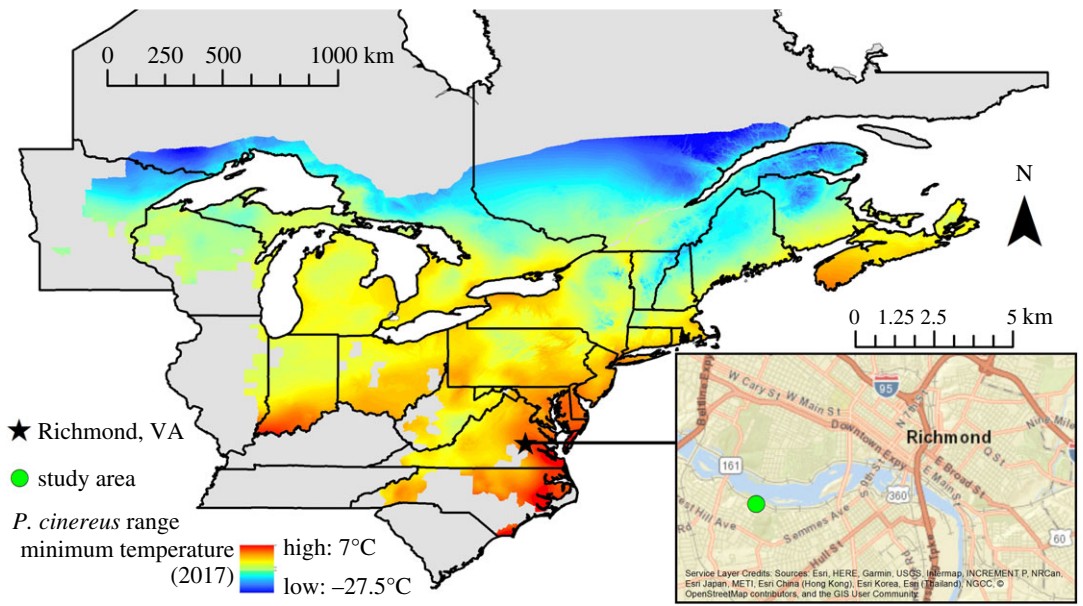

**Figure 1.** Map of the *P. cinereus* range (International Union for Conservation of Nature, Conservation International, and NatureServe 2014) symbolized by minimum surface temperature in 2017 [38] with deeper reds representing higher temperature and deeper blues representing lower temperature. The star indicates the location of Richmond, VA, USA and the inset shows the location of the study area (green circle). The range map shown is a compilation by the International Union for Conservation of Nature and may not reflect the current status of range edge populations or incorporate all historic populations.

cutaneous respiration. This species exhibits seasonal periods of surface activity where time periods that it can be detected at the soil surface are separated by long periods of surface inactivity where individuals retreat to underground refugia. Patterns of surface activity reflect behaviourally regulated maintenance of moisture and temperature requirements, where underground refugia are used to decrease risk of desiccation in warm weather and to decrease risk of freezing in cold weather. Thus, variation in phenology across the latitudinal range corresponds to variation in seasonal climate with some northern populations exhibiting two different seasons of surface activity in spring and autumn, and southern populations exhibiting a continuous season of surface activity with a period of inactivity during summer [39,40].

## 2.2. Study area

The study was conducted in the James River Park System, Richmond VA, USA (37°31′27 N, 77°28′29 W, elevation: 45 m; figure 1). This area was used as a rock quarry in the early 1900s and now serves as a public park characterized by secondary mixed hardwood deciduous forest. It is located between the James River and a suburban housing area with direct access for visitors and intrusions from roads and train tracks. The surrounding human population density is estimated at over 1000 persons per km$^2$ [41]. Temperature conditions in Richmond, VA are representative of the southern range edge of *P. cinereus*, with a mean annual minimum temperature of 3°C (figure 1, [26,38]). Other amphibian species identified in the study area include *Anaxyrus americanus*, *A. fowleri*, *Acris crepitans*, *Lithobates catesbeianus* and *L. sphenocephalus* [42], as well as *Ambystoma maculatum*.

## 2.3. Data collection

In November 2015, 5 × 10 m cover board arrays were established at three different sites (referred to as Sites 1, 2 and 3) in the study area, located at least 20 m apart from each other. Each array consisted of 50 uniquely identified pine cover boards (30.5 × 30.5 × 2.1 cm) spaced 1 m apart in a rectangular grid [43]. Each cover board served as a 'trap' that was surveyed on multiple occasions between autumn 2016 and February 2018 (Sites 1, 2 and 3 were surveyed on 10, 9 and 10 occasions, respectively). Each survey consisted of lifting each cover board and collecting all red-backed salamanders underneath. Collected salamanders were transported to laboratory facilities at University of Richmond, where they were measured and identified. On initial capture, individuals were given a unique mark by injecting

visual implant elastomer (Northwest Marine Technologies, Inc.) at up to four locations adjacent to each limb using combinations of up to five colours, a method found to be safe and reliable for individual identification in amphibians [44,45]. Within 24–48 h of collection, individuals were released under the same board where they were collected. Surveys were separated by at least two weeks to maximize cover board effectiveness and decrease behavioural responses to disturbance [46,47].

## 2.4. Spatial capture–recapture analysis

Our capture–mark–recapture protocol generated a binary encounter history for individuals observed at least once, where $y_{ijk}$ indicates whether or not individual $i$ was detected under board $j$ in occasion $k$. We analysed these data using spatial capture–recapture methods (SCR [48,49]) that jointly estimate the density of salamanders that visit the surface and spatially explicit detection probability as a decreasing function of the distance between the individuals' activity centres and the cover boards where they were captured [43,49]. These models reduce biases in estimates of abundance and density by accounting for individual movement and the resulting heterogeneity in detection probability due to the relationship between capture location and an individual's home range [50], overcoming previous methodological challenges for describing amphibian population abundances given the lack of control over detection probabilities.

The density model describes the distribution of individual activity centres, which are assumed to be distributed uniformly over a state-space $S$. The state-space $S$, which represents the area of interest defined by a sufficiently large buffer around the trapping array containing the activity centre of all individuals with non-negligible detection probabilities, is discretized at a resolution small enough to approximate continuous space relative to the species' movement and the trap arrangement. As described by Muñoz et al. [18,36] and Sutherland et al. [43], we defined $S$ as the centre points of a grid with a 4 m buffer around the cover board array and a $0.5 \times 0.5$ m grid cell resolution. Buffer selection was carried out by fitting the null SCR model with incrementally larger buffers ranging from 1 to 5.5 m until model parameters reached an asymptote as a function of buffer size. We selected the smallest buffer size (4 m) at the asymptote with state-space area of 189.25 m$^2$ to minimize computational demand while ensuring density estimates were not biased by the size of $S$.

The spatial detection model describes the detectability of an individual ($p$) as a function of the distance between trap locations ($x$) and the individual's activity centre ($s$):

$$y_{ijk}|s_i \sim \text{Bernoulli}(\,p[x_j, s_i])  \tag{2.1}$$

and

$$p[x_j, s_i] = p_0 \times \exp\left(\frac{d(x_j, s_i)^2}{(2\sigma^2)}\right),  \tag{2.2}$$

where $y_{ijk}$ is the binary observation, $s_i$ are the estimated individual activity centres, $x_j$ are the trap locations, $p_0$ is the baseline detection probability or the probability of detecting an individual at its activity centre, $d(x, s)$ is the Euclidean distance between trap $j$ and activity centre $s_i$, and $\sigma$ is the spatial scale parameter that characterizes the decline in detectability with distance from an activity centre. Given the implied model of space usage in the detection model in which $\sigma$ is the scale parameter of the half-normal distance function (equation (2.2)), we also estimated the 95% home range radius as $r_{0.95} = \sigma\sqrt{5.99}$ and its associated area as $A_{0.95} = \pi r^2$ [36,43]. Based on preliminary analyses that supported a strong behavioural response to capture events, we included a behavioural parameter in all models that allowed capture probability to change after the initial capture.

Based on the observed phenology, we assumed closure from autumn to spring. Each period of closure, referred to as a 'sampling session' (i.e. Sessions 1 and 2), consisted of four to six occasions and resulted in up to six opportunities to uniquely estimate each parameter (i.e. the combination of two sessions and three sites). To account for differences between sampling sessions and among the three sites sampled, we fitted a series of competing models that included session and/or site effects to explain variation in density $D$, baseline detection probability $p_0$ and the spatial scale parameter $\sigma$. To account for temporal variation in surface activity, we included the linear and quadratic effect of 'day of survey' as covariates on detectability. Considering all factor combinations among $D$, $p_0$ and $\sigma$ resulted in a total of 325 models. We analysed these models in R [51] using the package oSCR [52] and conducted AIC-based model selection following Arnold [53]. Encounter history data of red-backed salamanders, trap deployment data and codes for oSCR analysis are deposited in the Dryad Digital Repository (https://doi.org/10.5061/dryad.4bq41sg).

**Table 1.** Variability in density ($D$), baseline detection ($p_0$) and space use ($\sigma$) of the red-backed salamander as a function of site, session and time for the population in James River Park, Richmond, VA. Note: only top 10 ranked models presented. $X \times Y$ implies $X + Y + X:Y$. $Q_{day}$ denotes the quadratic effect of survey day (day + day²). $\Omega$ denotes model weights. Day is scaled (1 day = 10 days in calendar).

| density ($D$) | detection ($p_0$) | space use ($\sigma$) | AIC | $\Delta$AIC | $\Omega$ |
|---|---|---|---|---|---|
| ~site + session | ~site × session + $Q_{day}$ | ~site × session | 7694 | 0.00 | 0.36 |
| ~site + session | ~site + session + $Q_{day}$ | ~site × session | 7695 | 0.54 | 0.28 |
| ~site | ~site × session + $Q_{day}$ | ~site × session | 7697 | 2.51 | 0.10 |
| ~site × session | ~site + session + $Q_{day}$ | ~site × session | 7697 | 2.62 | 0.10 |
| ~site | ~site + session + $Q_{day}$ | ~site × session | 7698 | 3.37 | 0.07 |
| ~site × session | ~site × session + $Q_{day}$ | ~site × session | 7699 | 4.31 | 0.04 |
| ~site + session | ~day + session × site | ~site × session | 7701 | 6.83 | 0.01 |
| ~site × session | ~site + session + $Q_{day}$ | ~site + session | 7701 | 6.97 | 0.01 |
| ~site × session | ~day + session × site | ~site × session | 7703 | 8.73 | 0.00 |
| ~session | ~site × session + $Q_{day}$ | ~site × session | 7703 | 8.90 | 0.00 |

## 3. Results

Across both sampling sessions, a total of 863 individuals were captured at the three cover board arrays (326, 209 and 328 in Sites 1, 2 and 3, respectively). Of these, 653 individuals were captured more than once (226, 127 and 300, in Sites 1, 2 and 3, respectively). For a single individual, the maximum number of detections was 8. A total of 252 salamanders were captured at more than one board (90, 58 and 104, in Sites 1, 2 and 3, respectively) and the maximum number of boards where a single individual was observed was 6. For a given occasion, the maximum number of individuals under a single board was 8. Salamanders showed continuous surface activity from autumn to spring, retreating to underground refugia only during the summer (June–August).

We found support for session- and site-specific differences in salamander density, baseline detection and space use (cumulative model weight = 0.36; table 1; see electronic supplementary material for complete list of model weights). The estimated model coefficients for density ($\beta_D$), detection ($\beta_p$) and space use ($\beta_\sigma$) from the top model are presented in table 2. In Session 1, site-specific abundance and density ranged from 693 to 1184 salamanders and from 3.66 to 6.26 salamanders m$^{-2}$, respectively. In Session 2, site-specific salamander abundance and density ranged from 472 to 806 salamanders and from 2.49 to 4.26 salamanders m$^{-2}$, respectively (see table 3 for parameter estimates and confidence intervals).

Baseline detection probability varied by session and by site and there was support for a quadratic effect of survey day, indicating highest detectability at the 314th day of the year (November 10) and lower detection at the beginning of autumn and end of spring (table 1 and figure 2). In Session 1, maximum site-specific baseline detection was $p_{0_1} = 0.040$, $p_{0_2} = 0.021$ and $p_{0_3} = 0.042$, respectively. During Session 2, maximum site-specific baseline detection was $p_{0_1} = 0.015$, $p_{0_2} = 0.014$ and $p_{0_3} = 0.030$, respectively (figure 2). As was the case in preliminary analyses, *P. cinereus* detection increased after initial capture (positive behavioural response: $\beta_{p(b)} = 2.326$; table 2).

Similarly, the spatial scaling parameter $\sigma$ varied across sessions and sites (table 1 and figure 3). In Session 1, the estimated site-specific spatial scale parameter was $\sigma_1 = 0.95$ m, $\sigma_2 = 1.54$ m and $\sigma_3 = 1.30$ m, respectively (figure 3). In Session 2, the estimated site-specific spatial scale parameter was $\sigma_1 = 1.67$ m, $\sigma_2 = 1.44$ m and $\sigma_3 = 1.39$ m, respectively (figure 3). Site-specific 95% home range size varied from 17.1 to 44.5 m$^2$ in Session 1, and from 36.5 to 52.7 m$^2$ in Session 2 (table 3).

## 4. Discussion

Our analysis reveals extremely high densities and large home ranges in a suburban population of red-backed salamanders near its southern geographical range limit. We demonstrate that this suburban population follows the phenology expected from temperature limitations on surface activity in warm southern climates, and yet maintains higher abundances and spatial displacement during the active

**Table 2.** Regression coefficients of the top model describing the variability in density $D$, baseline detection ($p_0$), and space use ($\sigma$) of the red-backed salamander as a function of site, session and time. Note: day is scaled (1 day = 10 days in calendar).

| factors | slope | s.e. |
|---|---|---|
| **detection $p_0$** | | |
| $\beta_{p(intercept)}$ | −3.390 | 0.197 |
| $\beta_{p(site2)}$ | −0.646 | 0.215 |
| $\beta_{p(site3)}$ | 0.064 | 0.186 |
| $\beta_{p(session2)}$ | −1.000 | 0.257 |
| $\beta_{p(day)}$ | 0.055 | 0.027 |
| $\beta_{p(day2)}$ | −0.004 | 0.001 |
| $\beta_{p(site2*session2)}$ | 0.557 | 0.445 |
| $\beta_{p(site3*session2)}$ | 0.640 | 0.330 |
| $\beta_{p(b)}$ | 2.326 | 0.122 |
| **movement $\sigma$** | | |
| $\beta_{\sigma(intercept)}$ | −0.047 | 0.069 |
| $\beta_{\sigma(site2)}$ | 0.478 | 0.115 |
| $\beta_{\sigma(site3)}$ | 0.313 | 0.088 |
| $\beta_{\sigma(session2)}$ | 0.562 | 0.131 |
| $\beta_{\sigma(site2*session2)}$ | −0.627 | 0.213 |
| $\beta_{\sigma(site3*session2)}$ | −0.496 | 0.154 |
| **density $D$** | | |
| $\beta_{D(intercept)}$ | 0.447 | 0.102 |
| $\beta_{D(site2)}$ | −0.536 | 0.170 |
| $\beta_{D(site3)}$ | −0.398 | 0.130 |
| $\beta_{D(session2)}$ | −0.384 | 0.139 |

**Table 3.** Site-specific density of red-backed salamanders (individuals $m^{-2}$) and 95% home range size ($m^2$) of red-backed salamanders across sessions in Richmond, VA. 95% confidence intervals in parenthesis.

| site | density (95% CIs) salamanders $m^{-2}$ | 95% home range size (95% CIs) $m^2$ |
|---|---|---|
| **Session 1** | | |
| 1 | 6.26 (5.12, 7.64) | 17.1 (13.1, 22.4) |
| 2 | 3.66 (2.78, 4.82) | 44.5 (30.6, 64.8) |
| 3 | 4.20 (3.49, 5.06) | 32.0 (22.4, 40.7) |
| **Session 2** | | |
| 1 | 4.26 (3.19, 5.68) | 52.7 (32.8, 84.6) |
| 2 | 2.49 (1.74, 3.57) | 39.1 (19.8, 77.6) |
| 3 | 2.86 (2.21, 3.70) | 36.5 (24.6, 54.3) |

season relative to other populations across its range. This suggests that ecological factors that can negatively affect plethodontid populations (e.g. warm temperature) alone do not necessarily constrain population processes in southern red-backed salamanders.

The Richmond red-backed salamander population exhibited continuous surface activity from autumn to spring, retreating underground during the hot summer months (June–August). Although there is reduced activity during the coldest month (January), individuals were still observed at the surface. This follows the

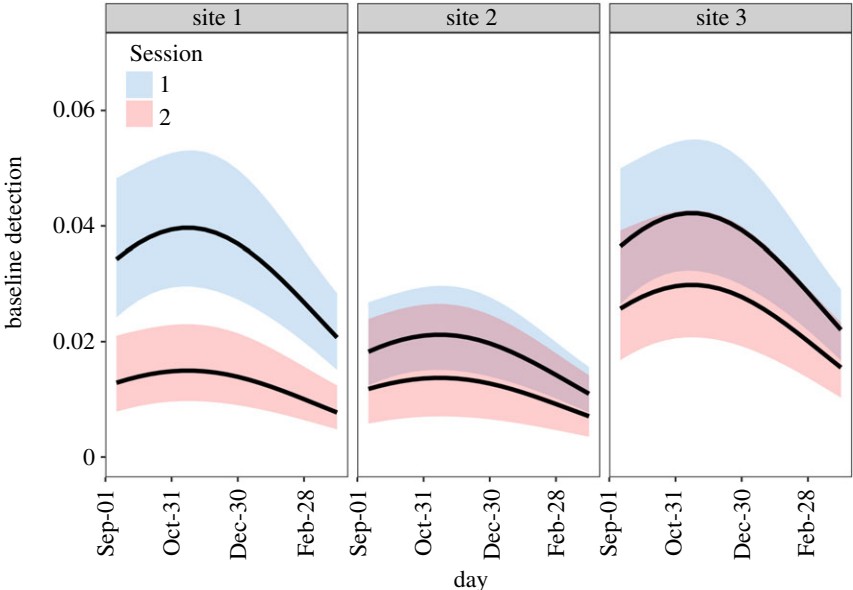

**Figure 2.** Site-specific baseline detection probability of the red-backed salamander at the southern edge range across sessions. Panel numbers represent sites. Lines and shaded areas represent model predictions and 95% CI, respectively.

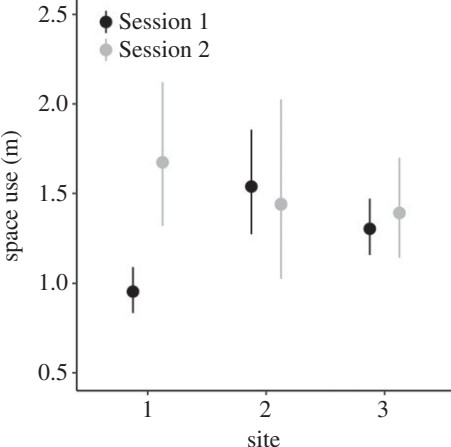

**Figure 3.** Site-specific space use (m) of red-backed salamanders at the southern range edge across sessions. Black and grey dots represent mean estimates for Sessions 1 and 2, respectively. Solid lines represent 95% CI.

observed phenology in surface activity of previously described southern populations in Tennessee [39]. It is also consistent with our SCR analysis showing that baseline detection probability follows a curvilinear relationship with time, where detection peaked during autumn and decreased at the onset and end of summer. Our results revealed complex dynamics where each population parameter assessed (i.e. density, baseline detection, space use) varied by both session and site. While answering questions regarding the mechanisms driving inter-annual variation will require longitudinal data, our analysis uses robust models to strongly suggest red-backed salamander populations are impacted by multiple localized factors that could create a mosaic of population responses to environmental constraints across latitudes, contradicting the expectation of low density and small home range size at the range edge [54].

Red-backed salamander densities reported here are higher than those reported in the literature across the range. On average, our site-specific densities ranged from 3.10 to 5.26 salamanders $m^{-2}$, while published estimates range from 0.05 to 3.28 salamanders $m^{-2}$ [27,55–64]. However, effectively comparing density estimates across the range is challenging because multiple methodologies have been used for both data collection and analysis (e.g. counts, traditional mark–recapture). A more direct comparison can be made with recently published spatial capture–recapture estimates of density and space use using the same cover board sampling design employed in this study. These studies report site-specific densities ranging

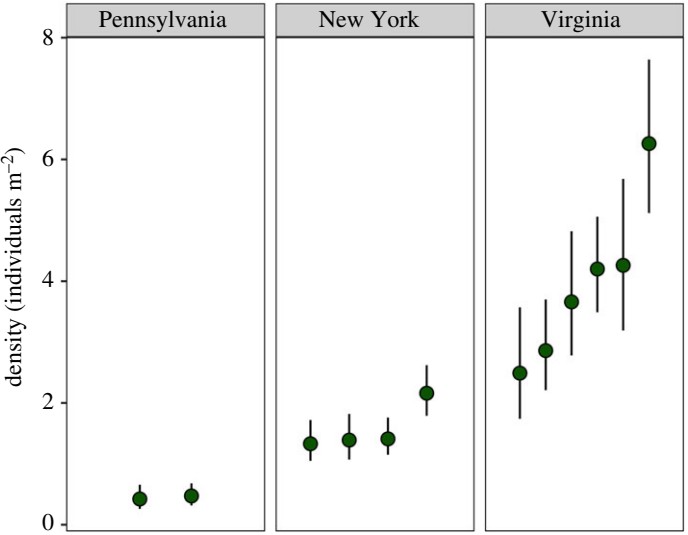

**Figure 4.** Comparison of SCR-based density estimates (individuals m$^{-2}$) for eastern red-backed salamander (*Plethodon cinereus*) across the habitat range. Density estimates are presented in increasing order of magnitude in density estimate for Pennsylvania [36], New York [43] and Virginia. The Pennsylvania estimates are from one site in spring and autumn of 2015, the New York estimates are from four sites in autumn 2014, and the Virginia estimates correspond to table 2. Solid black lines represent 95% CI.

from 0.42 to 0.47 m$^{-2}$ in Pennsylvania during spring and autumn seasons in 2015 (PA, [36]) and from 1.33 to 2.16 m$^{-2}$ in New York during autumn 2014 (NY, [43]; figure 4 presents a comparison between SCR published estimates). Similarly, estimated home ranges for *P. cinereus* in our study were large relative to PA and NY populations. As assessed by the mean 95% area of use, the average home range across sites for our population was 37.0 m$^2$ in contrast to mean home ranges of approximately 34.9 m$^2$ (95% area of use, [36]) and 16.28 m$^2$ (95% area of use, [43]). This is surprising given the limitations on physiological performance of *P. cinereus* at southern latitudes, the peripheral nature of the population relative to its geographical range, and the high level of disturbance surrounding the study area.

Southern red-backed salamander populations experience warmer environments that would be expected to constrain their population size and space use particularly given their ectothermic physiology, dependence on cutaneous respiration and susceptibility to desiccation. In warmer environments, *P. cinereus* is obligated to increase its metabolic expenditure [30,31], while its digestive performance and surface activity (e.g. foraging) becomes more limited [29,65]. Although this is expected to have direct demographic consequences (e.g. reduced population density), our analysis shows that southern temperatures alone do not limit red-backed salamander density and movement in our population and that other local factors (e.g. rainfall, genetic composition) may be compensating for the expected negative effects of temperature at the population level. For instance, density and baseline detection probability decreased in all sites by 32% and up to 63%, respectively, during the second session surveyed. This suggests that our population is not exempt from environmental pressure and that session-specific environmental effects (e.g. temperature) may be acting on the surface activity of individuals, and hence on the demographic and spatial processes of the population.

Density and home range size may interact with each other; i.e. density can reduce individual movement due to intraspecific competition [35]. The highly territorial behaviour of *P. cinereus* towards conspecifics suggests strong density-dependent effects over spatial processes such as movement and resulting home range size. However, our analysis does not support this pattern, suggesting home ranges are not necessarily limited by density in this population, but further examination of this pattern is warranted. Jaeger [58] hypothesized that seasonal changes in spatial distribution of *P. cinereus* were driven by intraspecific competition resulting from limited food and shelter rather than seasonal changes in density, which supports the lack of density dependence in space use we observed. By contrast, other studies have suggested that habitat settlement choice by *P. cinereus* is positively affected by the attraction to conspecifics, finding high aggregation of individuals during the active season of surface activity [66].

Our findings highlight the opportunity to identify and quantify environmental and biotic factors shaping terrestrial salamander population processes and test hypotheses on the role of selection in range edge populations. This is important as the response of species to changing environments is

expected to be determined mostly by population responses in these areas [17]. Isolated populations at low-latitude range edges can present unique genetic assemblages that, when grouped together, can hold the bulk of the species' genetic diversity with direct implications to conservation [67]. Future work should focus on implementing a standard robust analysis across the wide geographical range in order to quantify latitude-driven differences and understand factors explaining the observed extreme densities and large home ranges near the southern range limit.

Ethics. All procedures were approved by the Institutional Animal Care and Use Committee at the University of Richmond (Protocol 16-03-001). Research was conducted under scientific permits from the Virginia Department of Game and Inland Fisheries (nos. 056056 and 061682).

Data accessibility. Data available from the Dryad Digital Repository: https://doi.org/10.5061/dryad.4bq41sg [68].

Authors' contributions. R.H.-P. collected field data, carried out the spatial capture–recapture data analysis and statistical analysis, participated in the design of the study and drafted the manuscript. C.S. developed the statistical model, participated in data analysis and results interpretation and critically revised the manuscript. L.M.T. collected field data, participated in the design and coordination of the study, contributed to results interpretation and critically revised the manuscript. K.L.G. collected field data, led the design and coordination of the study, secured research funding, contributed to results interpretation and helped draft the manuscript.

Competing interests. Authors declare no competing interest.

Funding. This work was funded by the University of Richmond School of Arts and Sciences (CFD Post-doctoral Fellowship to R.H.-P.) and the Department of Biology.

Acknowledgements. We are grateful to students at the University of Richmond who contributed to fieldwork and mark–recapture data collection, particularly Sarah Timko, Christian Law, Alexis Porter, Khalea Sanchez, Amelia Tedesco and Maria Seitz. Special thanks to Nathan Burrell at the James River Park System, Ed Crawford and Spencer Bissett at Virginia Commonwealth University, and Jerry Gilfoyle and Fred Hagemeister at the University of Richmond for field and technical support. This project was conducted as part of the geographically distributed research and education network SPARCnet (Salamander Population Adaptation Research Collaboration Network) and we thank Evan Grant, David Miller, Sean Sterrett and David Muñoz for guidance and protocols.

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
