## [Reviewer comments · Royal Society Open Science]

Review History

RSOS-182192.R0 (Original submission)

Review form: Reviewer 1

Is the manuscript scientifically sound in its present form?

Yes

Are the interpretations and conclusions justified by the results?

Yes

Is the language acceptable?

Yes

Is it clear how to access all supporting data?

Yes

Do you have any ethical concerns with this paper?

No

Have you any concerns about statistical analyses in this paper?

No

Recommendation?

Major revision is needed (please make suggestions in comments)

Comments to the Author(s)

The ecology of southern salamanders differs from northern salamanders. Using state of the art spatial capture-recapture methods, the authors show that densities and home ranges are different. Thus, this study makes an important contribution to the growing body of studies which document intraspecific variation among amphibian populations.

I like the methods and results sections but I don't like the introduction and discussion.

1. Comments on the introduction:

The hypothesis is that density and home range should vary spatially. I don't think that the reasons given why home ranges should vary spatially are convincing (there is not much on line 103 ...). I think that home ranges should depend on resources such as food, refuges and mates. Why should it depend on the position within the species range? Is there a good theoretical justification for this hypothesis?

Line 49. I think it would be better to cite some data papers which quantify declines rather than general reviews. For North American amphibians, the paper by Grant et al. 2016 in Scientific Reports would be a good choice.

Line 61, 79. A nice example is the paper by Petrovan and Schmidt 2016 in Plos One.

Line 66-67. Less favourable habitats regulate amphibian populations. Isn't that circular reasoning? How do you infer habitat suitability? Is habitat less suitable because abundance is lower? Anyway, I think the sentence would be better if you would avoid the term 'regulation'. Regulation has a specific meaning in population ecology (see Sinclair, A. R. E. 1989. Population regulation in animals. - In: Cherrett, J. M. (ed.), Ecological concepts. Blackwell, pp. 197-240). (I would also avoid 'regulation' on lines 237 and 251.)

Line 67-70. The evidence listed for temperature as a factor regulating populations is very indirect. I would cite some papers which actually show how temperature affects vital rates (e.g. Reading 2007 in Oecologia; I'm sure there are similar papers for terrestrial salamanders).

Line 89-90. Amburgey et al. 2018 in Global Change Biology is a good paper on how position in the range affects amphibian population dynamics. There is also the abundance-center hypothesis which predicts abundance in relation to position in the range.

2. Comments on the discussion:

In my opinion, the discussion is too long. I think it should be shortened (50% is possible, I think; for example, I think the discussion of local adaptation is very speculative.) and focused on the key results. The result is that home ranges and densities are different from home ranges and densities elsewhere. That's an interesting result but a long discussion is not worthwhile if you cannot explain the causes. You would need a multi-site SCR study to test which factors affect density and home range size.

3. Other comments

I think you should define 'density' precisely. You estimate the density of the salamander population that is present at the surface? You don't estimate 'superpopulation' size which would also include salamanders which are underground?

Line 187. Only after initial capture or after every capture?

Reference [24] is missing (it's the one on local extinction due to stochastic events).

References 5 and 24 are identical.

Review form: Reviewer 2

Is the manuscript scientifically sound in its present form?

Yes

Are the interpretations and conclusions justified by the results?

Yes

Is the language acceptable?

Yes

Is it clear how to access all supporting data?

Yes

Do you have any ethical concerns with this paper?

No

Have you any concerns about statistical analyses in this paper?

No

Recommendation?

Accept with minor revision (please list in comments)

Comments to the Author(s)

This paper illustrates a small 2-year study on red-backed salamanders using spatial capture-recapture at a few sites located on the range edge of the species. The survey design and statistical approaches were excellent. The value of long-term and spatially replicated monitoring cannot be emphasized enough, and hopefully SPARCnet enables this work to continue (and expand).

My only criticism is that the narrative in the discussion tends to go a bit beyond the data and analysis provided. The paper is very well written and the material is interesting and relevant. Having season and site specific parameters makes for difficult interpretation when the mechanisms are not known (as the authors state on Line 249), but the general comparisons with other published estimates are compelling. The authors should consider keeping more focus on what was actually observed and estimated, and what it means. For example, the large season/year differences for sigma at site 1 that were not observed for sites 2 and 3. Are there ecological reasons site 1 would differ from sites 2 and 3? Finally, the site-year interactions and differences should not be treated as absolute evidence of ecological variation as there is no influence of sampling error.

Season is a confusing description here, as opposed to "session" or "year". When discussion of season 1 vs season 2 happens, it evokes phenological seasons when in reality the seasons are different years of sampling. Session is a common alternative if concerns about using "year" is that the closed sampling periods spanned multiple calendar years.

Figure 4 is portraying site-year combinations as if they are independent sites (with an arbitrary "site" number on the x-axis). Are the NY and PA data also site-year combos? If the true comparison is between regions, maybe a mean (+SE) density per region is necessitated. Having all the estimates together is great but some of that variability is sampling error, not true ecological differences.

Minor comments:

L73: It's not clear why anthropogenic disturbances and forest fragmentation would "play a role"

in range edge dynamics. They make sense as referenced in L76 (“together with the combined effect of climate change and human modifications”) as independent processes that interact with range edge processes. But range edges can occur anywhere, independent of human influences. Just needs some rephrasing here.

L158: “captured”

L172: Confusing to call an area measurement the buffer size. Better to list the buffer size and the resulting state space area.

L177: Subscripted indices (and Greek letters) should be italicized (or not) to be consistent with text. Italics are commonly used for such notation.

L178: The “e” here should probably be “exp”, unless the function is a superscript: $e^{\text{dist}(x,s)}$.

L181: List the terms in the order they appear in the equation.

L183: “...sigma is equivalent to space use” is awkward and vague. Sigma is the scale parameter of the half-normal distance function. The “implied model of space usage” is the distance function itself, not a single parameter.

L185: Size is vague. “Area” is the intended metric.

L325: Maybe a figure of density vs. sigma would be enlightening? Hard to see how a “significant” relationship could be illustrated with only 2 data points (conceivably site1-year1 vs. site1-year2). I think it would be useful to include reference to Murray Efford’s recent Ecography paper (2016) on density-dependence in home-range size, especially given the use of SCR modeling.

<https://doi.org/10.1111/ecog.01511>

L348: “The variation of population parameters among sites and site-season interactions we report demonstrates the dynamic properties of southern populations of red-backed salamanders.” Not sure such inferences can be supported from 6 data points.

Decision letter (RSOS-182192.R0)

24-Apr-2019

Dear Dr Hernández-Pacheco,

The editors assigned to your paper (“Unexpected spatial population ecology of a widespread terrestrial salamander near its southern range edge”) have now received comments from reviewers. We would like you to revise your paper in accordance with the referee and Associate Editor suggestions which can be found below (not including confidential reports to the Editor). Please note this decision does not guarantee eventual acceptance.

Please submit a copy of your revised paper before 17-May-2019. Please note that the revision deadline will expire at 00.00am on this date. If we do not hear from you within this time then it will be assumed that the paper has been withdrawn. In exceptional circumstances, extensions may be possible if agreed with the Editorial Office in advance. We do not allow multiple rounds of revision so we urge you to make every effort to fully address all of the comments at this stage. If deemed necessary by the Editors, your manuscript will be sent back to one or more of the original reviewers for assessment. If the original reviewers are not available, we may invite new reviewers.

To revise your manuscript, log into <http://mc.manuscriptcentral.com/rsos> and enter your Author Centre, where you will find your manuscript title listed under “Manuscripts with Decisions.” Under “Actions,” click on “Create a Revision.” Your manuscript number has been

appended to denote a revision. Revise your manuscript and upload a new version through your Author Centre.

- Data accessibility

If you wish to submit your supporting data or code to Dryad (<http://datadryad.org/>), or modify your current submission to dryad, please use the following link:
<http://datadryad.org/submit?journalID=RSOS&manu=RSOS-182192>

- Competing interests

- Authors' contributions

- Acknowledgements

- Funding statement

Kind regards,
Andrew Dunn
Royal Society Open Science
openscience@royalsociety.org

on behalf of Professor Len Thomas (Associate Editor) and Kevin Padian (Subject Editor)
openscience@royalsociety.org

Associate Editor's comments (Professor Len Thomas):

Associate Editor: 1

Comments to the Author:

Both reviewers are pretty positive about your work. One reviewer recommends some larger-scale editorial changes, particularly a shortening of the discussion section. Both reviewers note that you stray far from your actual findings and into the realm of speculation. I am therefore recommending major revision - please focus the paper, particularly the discussion, on what your data and analyses actually show. Please provide a point-by-point response showing what changes you have made to reviewers' comments.

Comments to Author:

Reviewers' Comments to Author:

Reviewer: 1

Comments to the Author(s)

The ecology of southern salamanders differs from northern salamanders. Using state of the art spatial capture-recapture methods, the authors show that densities and home ranges are different. Thus, this study makes an important contribution to the growing body of studies which document intraspecific variation among amphibian populations.

I like the methods and results sections but I don't like the introduction and discussion.

1. Comments on the introduction:

The hypothesis is that density and home range should vary spatially. I don't think that the reasons given why home ranges should vary spatially are convincing (there is not much on line 103 ...). I think that home ranges should depend on resources such as food, refuges and mates. Why should it depend on the position within the species range? Is there a good theoretical justification for this hypothesis?

Line 49. I think it would be better to cite some data papers which quantify declines rather than general reviews. For North American amphibians, the paper by Grant et al. 2016 in Scientific Reports would be a good choice.

Line 61, 79. A nice example is the paper by Petrovan and Schmidt 2016 in Plos One.

Line 66-67. Less favourable habitats regulate amphibian populations. Isn't that circular reasoning? How do you infer habitat suitability? Is habitat less suitable because abundance is lower? Anyway, I think the sentence would be better if you would avoid the term 'regulation'. Regulation has a specific meaning in population ecology (see Sinclair, A. R. E. 1989. Population regulation in animals. - In: Cherrett, J. M. (ed.), Ecological concepts. Blackwell, pp. 197-240). (I would also avoid 'regulation' on lines 237 and 251.)

Line 67-70. The evidence listed for temperature as a factor regulating populations is very indirect. I would cite some papers which actually show how temperature affects vital rates (e.g. Reading 2007 in Oecologia; I'm sure there are similar papers for terrestrial salamanders).

Line 89-90. Amburgey et al. 2018 in Global Change Biology is a good paper on how position in the range affects amphibian population dynamics. There is also the abundance-center hypothesis which predicts abundance in relation to position in the range.

2. Comments on the discussion:

In my opinion, the discussion is too long. I think it should be shortened (50% is possible, I think; for example, I think the discussion of local adaptation is very speculative.) and focused on the key results. The result is that home ranges and densities are different from home ranges and densities elsewhere. That's an interesting result but a long discussion is not worthwhile if you cannot explain the causes. You would need a multi-site SCR study to test which factors affect density and home range size.

3. Other comments

I think you should define 'density' precisely. You estimate the density of the salamander population that is present at the surface? You don't estimate 'superpopulation' size which would also include salamanders which are underground?

Line 187. Only after initial capture or after every capture?

Reference [24] is missing (it's the one on local extinction due to stochastic events).

References 5 and 24 are identical.

Reviewer: 2

Comments to the Author(s)

This paper illustrates a small 2-year study on red-backed salamanders using spatial capture-recapture at a few sites located on the range edge of the species. The survey design and statistical approaches were excellent. The value of long-term and spatially replicated monitoring cannot be emphasized enough, and hopefully SPARCnet enables this work to continue (and expand).

My only criticism is that the narrative in the discussion tends to go a bit beyond the data and analysis provided. The paper is very well written and the material is interesting and relevant. Having season and site specific parameters makes for difficult interpretation when the mechanisms are not known (as the authors state on Line 249), but the general comparisons with other published estimates are compelling. The authors should consider keeping more focus on what was actually observed and estimated, and what it means. For example, the large season/year differences for sigma at site 1 that were not observed for sites 2 and 3. Are there ecological reasons site 1 would differ from sites 2 and 3? Finally, the site-year interactions and differences should not be treated as absolute evidence of ecological variation as there is no influence of sampling error.

Season is a confusing description here, as opposed to “session” or “year”. When discussion of season 1 vs season 2 happens, it evokes phenological seasons when in reality the seasons are different years of sampling. Session is a common alternative if concerns about using “year” is that the closed sampling periods spanned multiple calendar years.

Figure 4 is portraying site-year combinations as if they are independent sites (with an arbitrary “site” number on the x-axis). Are the NY and PA data also site-year combos? If the true comparison is between regions, maybe a mean (+SE) density per region is necessitated. Having all the estimates together is great but some of that variability is sampling error, not true ecological differences.

Minor comments:

L73: It’s not clear why anthropogenic disturbances and forest fragmentation would “play a role” in range edge dynamics. They make sense as referenced in L76 (“together with the combined effect of climate change and human modifications”) as independent processes that interact with range edge processes. But range edges can occur anywhere, independent of human influences. Just needs some rephrasing here.

L158: “captured”

L172: Confusing to call an area measurement the buffer size. Better to list the buffer size and the resulting state space area.

L177: Subscripted indices (and Greek letters) should be italicized (or not) to be consistent with text. Italics are commonly used for such notation.

L178: The “e” here should probably be “exp”, unless the function is a superscript: $e^{\text{dist}(x,s)}$.

L181: List the terms in the order they appear in the equation.

L183: “...sigma is equivalent to space use” is awkward and vague. Sigma is the scale parameter of the half-normal distance function. The “implied model of space usage” is the distance function itself, not a single parameter.

L185: Size is vague. “Area” is the intended metric.

L325: Maybe a figure of density vs. sigma would be enlightening? Hard to see how a “significant” relationship could be illustrated with only 2 data points (conceivably site1-year1 vs. site1-year2). I think it would be useful to include reference to Murray Efford’s recent Ecography paper (2016) on density-dependence in home-range size, especially given the use of SCR modeling.

<https://doi.org/10.1111/ecog.01511>

L348: “The variation of population parameters among sites and site-season interactions we report demonstrates the dynamic properties of southern populations of red-backed salamanders.” Not sure such inferences can be supported from 6 data points.

Author's Response to Decision Letter for (RSOS-182192.R0)

See Appendix A.

Decision letter (RSOS-182192.R1)

21-May-2019

Dear Dr Hernández-Pacheco,

I am pleased to inform you that your manuscript entitled "Unexpected spatial population ecology of a widespread terrestrial salamander near its southern range edge" is now accepted for publication in Royal Society Open Science.

on behalf of Professor Len Thomas (Associate Editor) and Kevin Padian (Subject Editor)
openscience@royalsociety.org

Associate Editor Comments to Author (Professor Len Thomas):
Associate Editor

Comments to the Author:

Well done for a thorough revision, addressing all reviewers' concerns. Overall, a very nice contribution.

Follow Royal Society Publishing on Twitter: [@RSocPublishing](https://twitter.com/RSocPublishing)

Appendix A

Jeremy Sanders
Editor-in-chief
Royal Society Open Science

Thank you so much for the positive evaluation and for the invitation to submit a revised version of the attached manuscript entitled “Unexpected spatial population ecology of a widespread terrestrial salamander near its southern range edge”. We also like to thank the Associate Editor and the referees that helped improving significantly the manuscript by their excellent comments. We have addressed all the reviewers’ comments and suggestions in this revised version and hope that it fulfills the standard for being published in Royal Society Open Science. We have also deposited all of our data and codes under the following temporary Dryad doi; doi.org/10.5061/dryad.4bq41sg.

Please, find our detailed answers to each comment below.

Sincerely,
Raisa Hernandez Pacheco

Comments to Author:

Reviewers' Comments to Author:
Reviewer: 1

Comments to the Author(s)

The ecology of southern salamanders differs from northern salamanders. Using state of the art spatial capture-recapture methods, the authors show that densities and home ranges are different. Thus, this study makes an important contribution to the growing body of studies which document intraspecific variation among amphibian populations.

I like the methods and results sections but I don’t like the introduction and discussion.

1. Comments on the introduction:

The hypothesis is that density and home range should vary spatially. I don’t think that the reasons given why home ranges should vary spatially are convincing (there is not much on line 103 ...). I think that home ranges should depend on resources such as food, refuges and mates. Why should it depend on the position within the species range? Is there a good theoretical justification for this hypothesis?

- We have added two citations supporting our hypothesis (Efford et al. 2016 and Muñoz et al. 2016). Given the expected relationship between resource availability and home range size, population density is hypothesized to have an inverse relationship with home range size – a hypothesis that has not been widely addressed in amphibians due to the lack of spatial capture recapture analyses. We now highlight this in Lines 96-98 of the revised version.

Line 49. I think it would be better to cite some data papers which quantify declines rather than general reviews. For North American amphibians, the paper by Grant et al. 2016 in Scientific Reports would be a good choice.

- We have added two citations in Line 49 that address amphibian decline quantitatively (Grant et al. 2016 and Houlahan et al. 2000).

Line 61, 79. A nice example is the paper by Petrovan and Schmidt 2016 in Plos One.

- We now cite Petrovan and Schmidt (2016) in Lines 62 (originally line 61).

Line 66-67. Less favourable habitats regulate amphibian populations. Isn't that circular reasoning? How do you infer habitat suitability? Is habitat less suitable because abundance is lower? Anyway, I think the sentence would be better if you would avoid the term 'regulation'. Regulation has a specific meaning in population ecology (see Sinclair, A. R. E. 1989. Population regulation in animals. – In: Cherrett, J. M. (ed.), Ecological concepts. Blackwell, pp. 197–240). (I would also avoid 'regulation' on lines 237 and 251.)

- We understand the reviewer's concern and have changed the sentence In Lines 66-67 to “Environmental conditions in these areas are characterized by warmer temperatures [17], a factor expected to negatively affect the demography of amphibian populations.” (Lines 65-67 of revised version).
- We have also replaced “regulating” in Line 235 (originally line 237) and “regulated” in Line 248 (originally line 251) by “negatively affect”, and “impacted”, respectively.

Line 67-70. The evidence listed for temperature as a factor regulating populations is very indirect. I would cite some papers which actually show how temperature affects vital rates (e.g. Reading 2007 in Oecologia; I'm sure there are similar papers for terrestrial salamanders).

- We now focus the sentence in Lines 67-69 on the effects of temperature on terrestrial salamander demography and cite Muñoz et al. (2016) which addressed survival as a function of temperature in *Plethodon cinereus*, finding that warmer temperatures reduce individual growth with negative demographic consequences as size defines survival. We also cite Peterman and Semlitsch (2013) that addressed the decreased in recruitment (site occupancy of juveniles) as a function of temperature in *Plethodon albagula*.

Line 89-90. Amburgey et al. 2018 in Global Change Biology is a good paper on how position in the range affects amphibian population dynamics. There is also the abundance-center hypothesis which predicts abundance in relation to position in the range.

- We now cite Amburgey et al. (2017) and Nadeau and Urban (2019) in Line 81 (originally line 89).

2. Comments on the discussion:

In my opinion, the discussion is too long. I think it should be shortened (50% is possible, I think; for example, I think the discussion of local adaptation is very speculative.) and focused on the key results. The result is that home ranges and densities are different from home ranges and densities elsewhere. That's an interesting result but a long discussion is not worthwhile if you cannot explain the causes. You would need a multi-site SCR study to test which factors affect density and home range size.

- We have shortened the discussion considering the reviewer's suggestions. We eliminated Lines 286-288, 289-318, 333-338, 346-348 from the original version to avoid speculation on local adaptation.
- We have also incorporated minor changes to the Abstract and Introduction sections accordingly.

3. Other comments

I think you should define 'density' precisely. You estimate the density of the salamander population that is present at the surface? You don't estimate 'superpopulation' size which would also include salamanders which are underground?

- Our density estimate represents the population of salamanders from a plot that come to the surface at any point in the session. We have clarified the salamander density represented in our method in Line 150-151.

Line 187. Only after initial capture or after every capture?

- It is only after the initial capture. We changed the sentence in Line 182 to "...we included a behavioural parameter in all models that allowed capture probability to change after the initial capture."

Reference [24] is missing (it's the one on local extinction due to stochastic events).

- We have corrected the Reference section.

References 5 and 24 are identical.

- We have corrected the Reference section.

Reviewer: 2

Comments to the Author(s)

This paper illustrates a small 2-year study on red-backed salamanders using spatial capture-recapture at a few sites located on the range edge of the species. The survey design and statistical approaches were excellent. The value of long-term and spatially replicated monitoring cannot be emphasized enough, and hopefully SPARCnet enables this work to continue (and expand).

My only criticism is that the narrative in the discussion tends to go a bit beyond the data and analysis provided. The paper is very well written and the material is interesting and relevant. Having season and site specific parameters makes for difficult interpretation when the mechanisms are not known (as the authors state on Line 249), but the general comparisons with other published estimates are compelling. The authors should consider keeping more focus on what was actually observed and estimated, and what it means. For example, the large season/year differences for sigma at site 1 that were not observed for sites 2 and 3. Are there ecological reasons site 1 would differ from sites 2 and 3? Finally, the site-year interactions and differences should not be treated as absolute evidence of ecological variation as there is no influence of sampling error.

- We now present a more concise Discussion to avoid speculation, as detailed above in

response to the previous reviewer. We have also eliminated Lines 248-249 from original version to avoid using site-year interactions as absolute evidence of ecological variation.

- Unfortunately, we have no site-specific information that could explain the observed variation in sigma so we avoided more speculation.

Season is a confusing description here, as opposed to “session” or “year”. When discussion of season 1 vs season 2 happens, it evokes phenological seasons when in reality the seasons are different years of sampling. Session is a common alternative if concerns about using “year” is that the closed sampling periods spanned multiple calendar years.

- When referring to sampling year, we replaced “season” by “session” throughout the manuscript to avoid confusion.

Figure 4 is portraying site-year combinations as if they are independent sites (with an arbitrary “site” number on the x-axis). Are the NY and PA data also site-year combos? If the true comparison is between regions, maybe a mean (+SE) density per region is necessitated. Having all the estimates together is great but some of that variability is sampling error, not true ecological differences.

- The PA estimates are from one site in spring and fall of 2015, while the NY estimates are from four site in a single year (Fall 2014). We have added this wording to the figure legend so the reader is aware of what is being shown. This structure (4 sites, one season in NY vs 1 site two seasons in PA) means that we are not comfortable computing post hoc averages. Rather, we agree that being more explicit about what the estimates are, is necessary. See lines 258-261; 580-585, and updates in Figure 4 where we have amended the x-axis accordingly.

Minor comments:

L73: It’s not clear why anthropogenic disturbances and forest fragmentation would “play a role” in range edge dynamics. They make sense as referenced in L76 (“together with the combined effect of climate change and human modifications”) as independent processes that interact with range edge processes. But range edges can occur anywhere, independent of human influences. Just needs some rephrasing here.

- We understand the reviewer’s point. We have eliminated Lines 73-76 from original; version.

L158: “captured”

- Corrected

L172: Confusing to call an area measurement the buffer size. Better to list the buffer size and the resulting state space area.

- Done (Line 167).

L177: Subscripted indices (and Greek letters) should be italicized (or not) to be consistent with text. Italics are commonly used for such notation.

- Done.

L178: The “e” here should probably be “exp”, unless the function is a superscript: $e^{\text{dist}(x,s)}$.

- Done (Line 173).

L181: List the terms in the order they appear in the equation.

- Done (Line 175).

L183: "...sigma is equivalent to space use" is awkward and vague. Sigma is the scale parameter of the half-normal distance function. The "implied model of space usage" is the distance function itself, not a single parameter.

- We have changed the sentence to "Given the implied model of space usage in the detection model in which σ is the scale parameter of the half-normal distance function..." (Line 178).

L185: Size is vague. "Area" is the intended metric.

- We changed "size" to "area" (Line 180).

L325: Maybe a figure of density vs. sigma would be enlightening? Hard to see how a "significant" relationship could be illustrated with only 2 data points (conceivably site1-year1 vs. site1-year2). I think it would be useful to include reference to Murray Efford's recent Ecography paper (2016) on density-dependence in home-range size, especially given the use of SCR modeling. <https://doi.org/10.1111/ecog.01511>

- We understand the reviewer's concern and we have change the sentence in Line 325 in original version to "However, our analysis do not support this pattern, suggesting home ranges are not necessarily limited by density in this population but further examination of this pattern is warranted." (Lines 285-287).
- We have included Efford et al. 2016 (see comments to first reviewer)

L348: "The variation of population parameters among sites and site-season interactions we report demonstrates the dynamic properties of southern populations of red-backed salamanders." Not sure such inferences can be supported from 6 data points.

- We understand the reviewer's concern and have eliminated such statement from the original version.